# Thermally Conductive and Electrical Insulation BNNS/CNF Aerogel Nano-Paper

**DOI:** 10.3390/polym11040660

**Published:** 2019-04-10

**Authors:** Xiu Wang, Zhihuai Yu, Huiyang Bian, Weibing Wu, Huining Xiao, Hongqi Dai

**Affiliations:** 1Jiangsu Co-Innovation Center of Efficient Processing and Utilization of Forest Resources, Nanjing Forestry University, Nanjing 210037, China; 18362985880@163.com (X.W.); yuzhihuai15@163.com (Z.Y.); hybian1992@njfu.edu.cn (H.B.); wbwu@njfu.edu.cn (W.W.); 2Department of Chemical Engineering, University of New Brunswick, Fredericton, NB E3B5A3, Canada

**Keywords:** BNNS, CNF, aerogel, 3D skeleton, nano-paper, thermal conductivity, electrical insulation

## Abstract

Adding heat conducting particles to a polymer matrix to prepare thermally conductive and electrical insulation materials is an effective approach to address the safety issues arising from the accumulation of heat in the working process of electronic devices. In this work, thermally conductive and electrical insulation nano-paper, consisting of Boron Nitride nano-sheet (BNNS) and cellulose nanofiber (CNF), was prepared using an aerogel 3D skeleton template method. For comparison, BNNS/CNF nano-paper was also produced using a simple blending method. At a BNNS loading of 50 wt%, the thermal conductivity of BNNS/CNF aerogel nano-paper and blended nano-paper at 70 °C are 2.4 W/mK and 1.2 W/mK respectively, revealing an increase of 94.4%. Under similar conditions, the volume resistivity of BNNS/CNF aerogel nano-paper and blended nano-paper are 4.0 × 10^14^ and 4.2 × 10^14^ Ω·cm respectively. In view of its excellent thermal conductivity and electrical insulation performance, therefore, BNNS/CNF aerogel nano-paper holds great potential for electronic-related applications.

## 1. Introduction

Advancements toward miniaturization and increasing-power in electronic devices, with their resulting continuous accumulation of internal heat, shortens the devices’ lives and causes safety issues such as explosions [1,2,3]. Dealing with the heat generated by electronic devices and improving their heat dispersion has become the focus of the electronic industry. Traditional polymer materials, the principal insulation material of electronic devices [4,5,6,7,8,9], typically have low thermal conductivity of around 0.1–0.5 W/mK at room temperature [10,11,12,13,14,15,16]. This is not enough to provide sufficient heat-dissipation requirements for electronic devices. Thermally conductive composite material with excellent insulation properties can be prepared by adding heat conducting filler particles to the polymer [17,18]. Boron Nitride nano-sheet (BNNS), exfoliated from hexagonal Boron Nitride (h-BN), has a similar crystal structure to graphene [19,20], which is a typical 2D thin sheet material [21,22,23]. It is an ideal filler for the preparation of thermal conductive nano-composites due to its high thermal conductivity, resistivity, thermal stability and low thermal expansion rate [24,25]. The key to increasing the composites’ thermal conductivity was to make the filler form a network of thermally conductive pathways [24,26]. Additionally, with the emergence of wearable electronic products in recent years, electronic packaging materials are required to be capable of bending and folding [27]. Cellulose nanofiber (CNF) has high yield, eco-friendly, and natural degradation properties [28], and CNF nano-paper has excellent strength and resistivity, and exhibits low thermal expansion coefficient, which is an ideal base for the preparation of a new flexible electronic substrate [29,30].

In this study, using CNF as the substrate, BNNS/CNF composite nano-paper with improved thermally conductivity was prepared using an aerogel 3D skeleton template method. For the purpose of comparison, a simple blending method was also applied. The resulting effects of BNNS content and distribution on the thermal conductivity and electrical insulation properties of BNNS/CNF composite nano-paper were systematically studied.

## 2. Materials and Methods

### 2.1. Materials and Chemicals

The cellulose used in this work was bleached softwood pulp (llim Pulp Co. Ltd., Bratsk, Russia). The hexagonal Boron Nitride (h-BN) powders with an average diameter of 1~2 µm were obtained from Aladdin reagent Co. (Shanghai, China). Isopropyl alcohol was supplied from Nanjing Chemical reagent Co. (Nanjing, China). Polyvinylidene Fluoride membrane (PVDF membrane) with a pore size of 0.22 µm was obtained from Aladdin reagent Co. (Shanghai, China).

### 2.2. Preparation of BNNS/CNF Nano-Paper

CNF was prepared according to previous work [31,32]. Five grams of bleached softwood pulp with mechanical pretreatment was suspended in 450 mL of deionized water and 50 mL of NaClO. Then, 0.08 g of TEMPO and 0.8 g of NaBr were added to the solution and stirred at 250 rpm at 25 °C for 8 h. The pH was adjusted to about 10.0–10.5 using HCl (20% *v*/*v*) and NaOH (0.5 mol/L). The reaction was then terminated by adding 10 mL of ethyl alcohol, followed by adjusting pH to 7 using HCl (1 mol/L). The suspension was dialyzed against deionized and distilled water using a dialysis bag with MW-CO 12,000~14,000 (D) after centrifuging at 5000 rpm 3 times. Finally, the suspension was treated with a homogenizer (FB-110X, ShangHai LiTu Mechanical Equipment Engineering Co. Ltd., Shanghai, China) under a pressure of 600 bar for 10 cycles. The liquid phase exfoliation of h-BN was prepared using methods previously utilized others [1]. A certain amount of h-BN powders was added into a mixture solvent of isopropanol and deionized water (1/1). The suspension was sonicated in X0-650 model sonication bath (xianou Instruments Ltd., Nanjing, China) for 8 h with a frequency of 200 kHz. The resulting dispersions were centrifuged using a TGL-15B supercentrifuge (Shanghai MeiYingPu Instrument Manufacturing Co., Ltd., Shanghai, China) at 8000 rpm for 20 min to remove nonexfoliated h-BN. The supernatants contained the BNNS.

BNNS/CNF nano-papers were prepared using blending and aerogel template methods. BNNS/CNF blended nano-paper was prepared via simple mixing. In brief, a certain amount of CNF was stirred with the BNNS dispersion for 30 min, the resulting solid content of the mixture being 0.5 wt%. The mixture was then transferred into an ice bath equipped with an ultrasonic mixer (X0-650, Xianou Instruments Ltd., Nanjing, China) which was operated at 300 W output power for 10 min. The mixture was poured into the filtering device (55 mm diameter) under 0.1 MPa of a suction filter until the bottom of the funnel was no longer dripping water. A Polyvinylidene Fluoride (PVDF) membrane with a pore size of 0.22 μm was used to filter film in the filtering device. The wet nano-paper was then transferred to the dryer of a Rapid Köthen sheet former (RK-2A PTI, Vorchdorf, Austria) under 0.5 MPa vacuum pressure. From this, the nano-paper was produced. All the samples were conditioned at 25 °C and 50% Relative Humidity (RH) for 72 h before measurements were taken. The BNNS/CNF aerogel nano-paper was prepared using an aerogel 3D skeleton template method. The BNNS solution was uniformly mixed with the CNF, keeping the solid content ratio of the BNNS to CNF at 3:1 and the concentration of the mixed solution at 0.5 wt%. The mixture was poured into polystyrene petri dishes (diameter 55 mm) and freezed in liquid nitrogen. The BNNS/CNF aerogel was completed after freeze-drying under −80 °C, 15 Pa for 72 h in a FD-1C-80 freeze dryer (Shanghai Yuming Instruments Ltd., Shanghai, China). The above BNNS/CNF aerogel was created in CNF suspension with solid content of 1.5 wt%, and transferred into a blast oven at 30 °C for several hours to produce the gel. Finally, the gel was transferred to the dryer of the Rapid Köthen sheet former (RK-2A PTI, Vorchdorf, Austria) under 0.5 MPa vacuum pressure. From this, the BNNS/CNF aerogel nano-paper was produced. All the samples were conditioned at 25 °C and 50% RH for 72 h before measurements were taken.

### 2.3. Characterization

The morphology of CNF was characterized using a Dimension Edge Atomic Force Microscope (AFM) (Bruker, Karlsruhe, Germany) in tapping mode at 300 kHz. The morphologies of BNNS and BNNS/CNF aerogel were observed using an environmental Scanning Electron Microscope (SEM) (Quanta-200, FEI, Hillsboro, OR, USA). The chemical characteristics of BNNS/CNF aerogel nano-paper were observed with an IRPrestige-21 Fourier Transform Infrared Spectrometer (FTIR) (Shimadzu Company, Kyoto, Japan). The crystal structure of BNNS/CNF aerogel nano-paper was determined using an Ultima IV X-ray diffraction (XRD) (Rigaku Corporation, Tokyo, Japan). The thermal stability of the BNNS/CNF aerogel nano-paper was measured via thermogravimetric (TG) analysis using a thermogravimetric analyzer (TGA) (Q5000IR, TA instruments, New Castle, DE, USA). The volume resistance was tested using an insulation resistance tester (ZC36, Shanghai Jingke Industrial Co., Shanghai, China) at 25 °C and 50% RH.

The thermal diffusion coefficient was measured under the N_2_ atmosphere of 50 mL/min using the laser flash technique (LFA 467 HT HyperFlash, NanoFlash, Netzsch, Seelze, Germany). The testing principle is shown in Figure 1. Every film was cut into a square with size of 10 mm × 10 mm. At the preset temperature, a pulse of light was transmitted instantaneously by the laser source and irradiated on the surface of the sample uniformly. The surface temperature was raised after absorbed light energy, and the energy was transmitted to the cold end. The infrared detector was used to continuously measure the corresponding temperature rise process of the central part of the upper surface of the sample. The curve of temperature as a function of time was obtained. The thermal diffusivity was obtained from equation: α = 0.1388 × d^2^/t_1/2_, in which d was the thickness of the sample and t_1/2_ was semi-temperature rising time, as exhibited in Figure 1. The specific heat of the sample was obtained by DSC 204 F1 Phoenix (Netzsch, Seelze, Germany), the density was measured using a real density meter with exhaust method. Thermal conductivity λ (W/mK) was calculated as a multiplication of density (ρ, g/cm^3^), specific heat (Cp, J/gK), and thermal diffusivity (α, mm^2^/s). Namely: λ(T) = α(T) × Cp(T) × ρ(T).

## 3. Results and Discussion

### 3.1. Morphologies of CNF and BNNS

Figure 2a shows the AFM image of CNF, and Figure 2b the corresponding height distribution measured from AFM topographic data of CNF from Figure 2a. The length of CNF was 300–1000 nm and its diameter was primarily in the range of 6–7 nm, as observed by the Tyndall effect in Appendix A. In order to improve the heat transmission efficiency of filler, in theory, filler particles should have greater contact area between them. Therefore, it was more advantageous to construct the network structure of phonon transfer as a heat carrier to reduce phonon scattering on the filler surface [21,25,33]. In this case, the ideal h-BN filler should have a larger specific surface area and reduced thickness. Studies have indicated that the thermal conductivity of BNNS at room temperature was 360 W/mK, close to the intrinsic thermal conductivity of h-BN base plane (001) [1]. Hence, BNNS with a thickness of only a few layers was more suitable as a filler of thermal conductive composites [34]. In this study, commercial h-BN was used as raw material for ultrasonic treatment in the mixture solvent of isopropanol and deionized water (1/1), and BNNS was produced using centrifugal separation, which could be dispersed into water to create stable suspension without precipitation for a long time period. Appendix A presents an optical photograph of BNNS solution and Tyndall effect when irradiated by a red laser beam. Figure 2c,d show the SEM and AFM images of BNNS, respectively. It can be seen that the thickness of BNNS was about 1 to 3 layers and transverse dimensions were about 500~1000 nm, which was thinner than commercial h-BN (Appendix A), indicating that the h-BN had successfully exfoliated.

### 3.2. Morphologies of BNNS/CNF Aerogel

The scheme of preparation for BNNS/CNF nano-papers are shown in Figure 3a. Figure 3b shows the optical photos of BNNS/CNF mixing solution (top) and BNNS/CNF aerogel (middle and bottom). The diameter of the aerogel was 55 mm with a thickness of approximately 7 mm. Figure 3c,d are the SEM images of BNNS/CNF aerogel. It can be seen that BNNS/CNF aerogel had a typical “honeycomb” structure. BNNS self-assembled on the surface of CNF and gradually accumulated along the fiber skeleton to form a 3D network structure. The desired contact area between the BNNS was maintained sufficiently high for the thermal resistance to be effectively reduced when heat was transferred along the BNNS/CNF frame. It was difficult to observe the CNF in SEM images because the proportion of CNF in the aerogel was low, and the size of CNF was relatively small, compared with BNNS, which was easily sheltered by BNNS. The surface morphology of BNNS/CNF aerogel nano-paper is shown in Appendix A, indicating that BNNS was distributed uniformly. Appendix A presents proper bending and folding properties of the BNNS/CNF aerogel nano-paper.

### 3.3. Chemical Characteristics and Thermal Stability of BNNS/CNF Aerogel Nano-Paper

Figure 4a,b show the FTIR spectra and XRD patterns of BNNS/CNF aerogel nano-paper. The absorption bands appeared in the region of 3400 cm^−1^ and 2900 cm^−1^ and were related to O–H stretching vibration and C–H stretching vibrations, respectively. Additionally, absorption bands caused by –C–O–C– were observed near 1100 cm^−1^, 1050 cm^−1^ and 990 cm^−1^. These are typical absorption bands of cellulose [28] which can be found in all samples. New strong peaks can be clearly observed at 800 cm^−1^ due to the introduction of BNNS, caused by B–N skeleton vibration [35]. As shown in Figure 4b, cellulose characteristic peaks were observed at 16.5° and 22.5° in XRD patterns, which represented (110) and (200) crystalline surfaces of cellulose [28], respectively. After adding BNNS, six new characteristic peaks belonging to BNNS surface were observed [36]. The absorption bands of cellulose in aerogel nano-paper were not displaced. The results of FTIR and XRD analysis indicated that there was no chemical reaction between BNNS and CNF in the mixing process and the chemical structures of both were preserved.

Figure 4c,d show the TG and DTG curves of BNNS/CNF aerogel nano-paper. Compared with the BNNS/CNF nano-paper, pure CNF nano-paper had the lower onset degradation temperature about 175 °C, owing to the carboxyl groups on the surface of CNF [37]. As can be seen in Figure 4c,d, the initial decomposition temperature and the maximum decomposition rate of the nano-paper gradually increased with the increase of BNNS loading in the nano-paper. It has been reported that pure h-BN exhibits high thermal stability on heating up to 800 °C under N_2_ atmosphere [38], suggesting that h-BN can effectively slow down the decomposition rate of nano-paper.

### 3.4. Thermal Conductivity and Electrical Properties of h-BN/CNF Composite Films

As shown in Figure 5a, the thermal conductivity of CNF nano-paper was 0.41 W/mK at 20 °C, which was lower than that of BNNS/CNF nano-paper. The thermal conductivity of the nano-paper prepared by both methods was increased by increasing the BNNS loading, implying that BNNS play a dominant role in enhancing the thermal conductivity of the nano-paper. For the same BNNS loading, aerogel nano-paper had higher thermal conductivity than blended nano-paper. This could be attributed to the thermally conductive pathways inside the aerogel nano-paper created by using the aerogel template method, which facilitated the propagation of phonons and improved thermal conductivity. Figure 5b shows the thermal conductivity of the nano-papers at different temperatures. By increasing their temperature, the thermal conductivities of aerogel nano-paper and blended nano-paper were increased. Such phenomenon may originate from the thermal expansion of CNF with the temperature increase, which gives larger pressure on 3D BNNS skeleton network at high temperature, resulting in close contact among the BNNS skeleton and reducing the interfacial thermal resistance among BNNS.

Secondly, when the temperature increased, the thermal expansion of CNF occurred, which will exert certain pressure on the BNNS thermally conductive pathways, making the filler contact closer and reducing the interface thermal resistance. At the same temperature, the thermal conductivity of aerogel nano-paper was always higher than that of blended nano-paper. Figure 5c shows the influence of BNNS loading on increments of thermal conductivity of the composites prepared by both methods at 20 °C, the reference for comparison being pure CNF nano-paper. The increments from the aerogel template method were clearly much higher than those from the blending method. This further demonstrates that an aerogel template method can form more thermally conductive pathways, thus improving the thermal conductivity of the composites.

Figure 5d presents the volume resistivity (Φ) of BNNS/CNF composite nano-paper with different BNNS loadings. Compared to pure CNF nano-paper, the volume resistivity of nano-papers increased with the increasing of BNNS loading. Moreover, blended nano-paper had higher volume resistivity than aerogel nano-paper at the same content of BNNS, which might be attributed to more evenly dispersed BNNS in nano-paper prepared over the blending process. Obviously, both the BNNS/CNF nano-paper obtained from different methods can fully meet the insulation requirements (Φ > 10^9^ Ω·cm) [39] of the packaging materials under the low voltage and low current working conditions of the electronics.

## 4. Conclusions

Thermal conductive and electrical insulation nano-paper was successfully prepared with BNNS and CNF through the aerogel 3D skeleton template method. The thermal conductivity of aerogel nano-paper was much higher than that of blended nano-paper. The difference in thermal conductivity was not significantly affected by variations in temperature. Even at 100 °C, the thermal conductivity of aerogel nano-paper remained significantly higher than that of blended nano-paper. Meanwhile, the volume resistivity of BNNS/CNF nano-paper was higher than that of pure CNF nano-paper, regardless of the manner used for preparing BNNS/CNF composites. Overall, BNNS/CNF aerogel nano-paper has excellent balanced thermal conductivity and electrical insulation performance, and holds great potential for a multitude of applications associated with electronics.

## Figures and Tables

**Figure 1 polymers-11-00660-f001:**
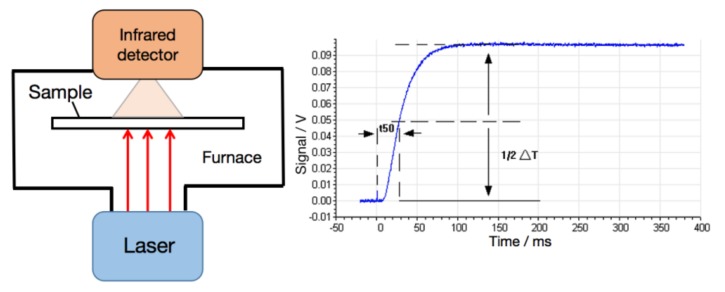
The principle of testing thermal diffusivity.

**Figure 2 polymers-11-00660-f002:**
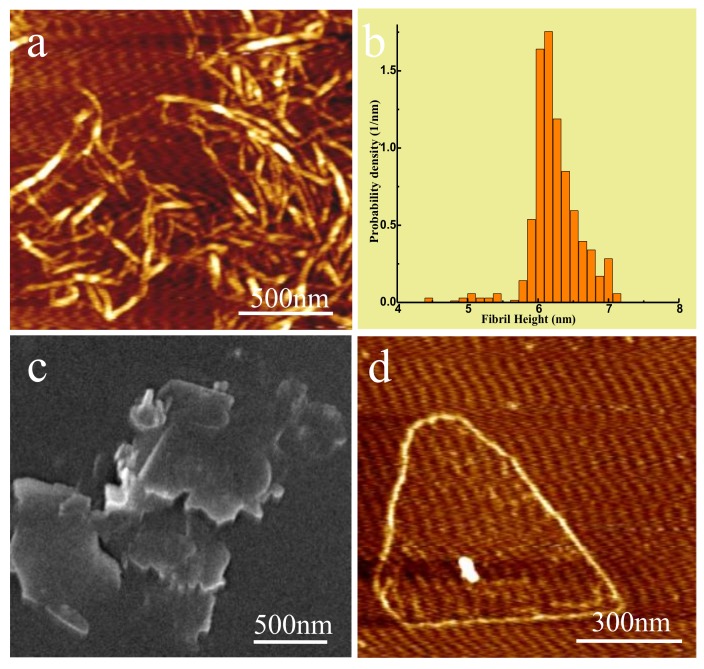
(**a**) AFM image of CNF, (**b**) The corresponding height distribution measured from AFM topographic data of CNF, (**c**) SEM Image of BNNS, (**d**) AFM image of BNNS.

**Figure 3 polymers-11-00660-f003:**
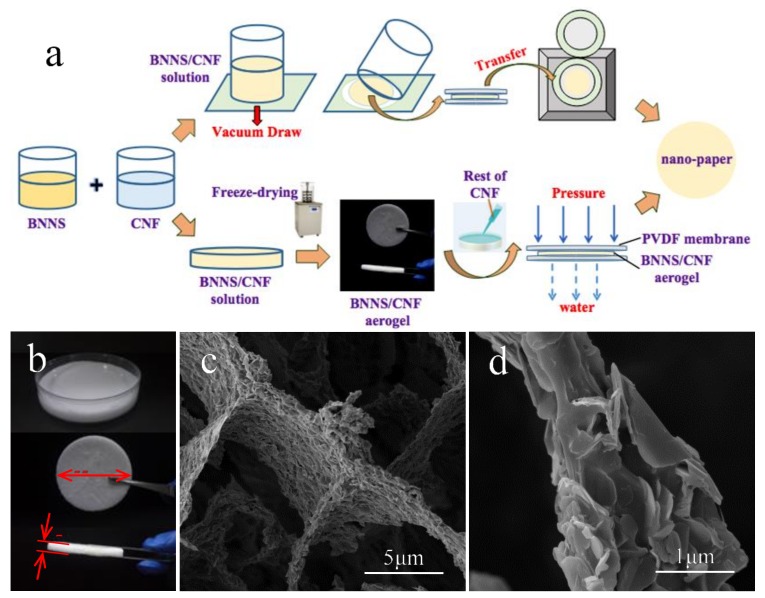
(**a**) Scheme of preparation of BNNS/CNF nano-papers, (**b**) Images of BNNS/CNF mixing solution and aerogel, (**c**) and (**d**) SEM images under different magnifications of BNNS/CNF aerogel.

**Figure 4 polymers-11-00660-f004:**
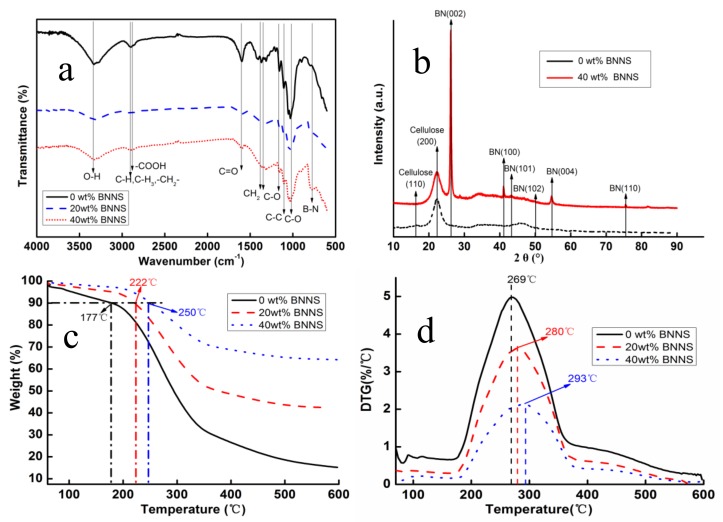
(**a**) FTIR spectra, (**b**) XRD patterns, (**c**) TG curves, (**d**) DTG curves of aerogel nano-papers.

**Figure 5 polymers-11-00660-f005:**
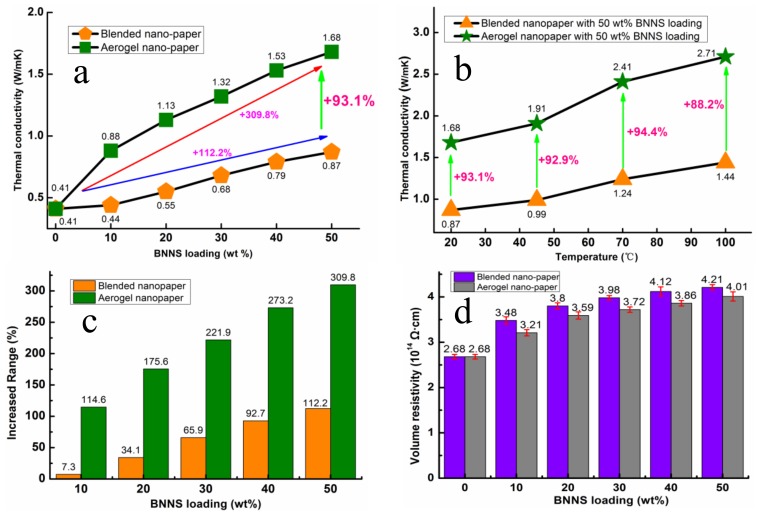
(**a**) Thermal conductivity of BNNS/CNF nano-paper, (**b**) The effects of temperature on BNNS/CNF nano-paper, (**c**) Increased range of thermal conductivity at 20 °C, (**d**) The effects of BNNS loading on volume resistivity of nano-paper.

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
