# Peer review of "Thermally Conductive and Electrical Insulation BNNS/CNF Aerogel Nano-Paper"

_polymers, 2019, doi:10.3390/polym11040660_

Reviewer 1 Report

1) I do not think that (given the accuracy of the measurement technique) it makes sense to indicate three digits after the decimal point like in the present manuscript: “2.403 W·m-1K-1 and 1.236 W·m-1K-1”. It should be rounded to 2.4 W/mK, etc. The same applies everywhere in the text.

2)The authors wrote: “Boron nitride nano-sheet (BNNS) exfoliated from hexagonal boron nitride (h-BN) has a similar crystal structure to graphene [….].” There have been prior works on h-BN and graphene as fillers in various thermal interface materials. These relevant papers should be cited in the introduction and the results compared when possible. The relevant literature include: F. Kargar, et al., "Thermal percolation threshold and thermal properties of composites with high loading of graphene and boron nitride fillers," ACS Appl. Mater. Interfaces, vol. 10, no. 43, pp. 37555–37565, 2018; Adv. Electron. Mater., pp. 1800558, 2018. DOI: 10.1002/aelm.201800558; K. M. F. Shahil et al., Solid State Commun., vol. 152, no. 15, pp. 1331–1340, 2012; J. D. Renteria, et al., “Strongly anisotropic thermal conductivity of free-standing...,” Adv. Funct. Mater., vol. 25, no. 29, pp. 4664–4672, 2015.

3) Provide details of the thermal conductivity measurements. Presently they are missing. How the mass density was determined? I am guessing one needs it if it was thermal diffusivity which was actually measured.

Author Response

Point 1. I do not think that (given the accuracy of the measurement technique) it makes sense to indicate three digits after the decimal point like in the present manuscript: “2.403 W·m-1K-1 and 1.236 W·m-1K-1”. It should be rounded to 2.4 W/mK, etc. The same applies everywhere in the text. 

Response 1: Thanks for the suggestion. The revision has been made over the entire manuscript; and two decimals were kept at the most. For example, the thermal conductivities at 0.413 W/mK and 0.437 W/mK have been changed to 0.41 W/mK and 0.44 W/mK.

Point 2. The authors wrote: “Boron nitride nano-sheet (BNNS) exfoliated from hexagonal boron nitride (h-BN) has a similar crystal structure to graphene [….].” There have been prior works on h-BN and graphene as fillers in various thermal interface materials. These relevant papers should be cited in the introduction and the results compared when possible. The relevant literature include: F. Kargar, et al., "Thermal percolation threshold and thermal properties of composites with high loading of graphene and boron nitride fillers," ACS Appl. Mater. Interfaces, vol. 10, no. 43, pp. 37555–37565, 2018; Adv. Electron. Mater., pp. 1800558, 2018. DOI: 10.1002/aelm.201800558; K. M. F. Shahil et al., Solid State Commun., vol. 152, no. 15, pp. 1331–1340, 2012; J. D. Renteria, et al., “Strongly anisotropic thermal conductivity of free-standing...,” Adv. Funct. Mater., vol. 25, no. 29, pp. 4664–4672, 2015.

Response 2: The advice of adding references is of great help to the improvement of the article. We have cited the relevant papers in the revised manuscript. In our experiment, the BNNS contents were 0 wt%, 10 wt%, 20 wt%, 30 wt%, 40 wt%, and 50 wt%, the test temperatures were 20 °C, 45 °C, 70 °C and 100 °C, and the thermal conductivities were through-plane. In these four references, boron nitride only used in one paper, the BNNS contents and test temperature also different, so it might not be suitable for comparison.

Point 3. Provide details of the thermal conductivity measurements. Presently they are missing. How the mass density was determined? I am guessing one needs it if it was thermal diffusivity which was actually measured.

Response 3: 1) We have added the details of the thermal conductivity measurements which have been marked in the revised manuscript: The testing principle is shown in Figure 1. Every film was cut into a square with a size of 10 mm × 10 mm. At the preset temperature, a beam of light pulse was transmitted instantaneously by the laser source and irradiated on the surface of the sample uniformly. The surface temperature was raised after absorbed light energy, and the energy was transmitted to the cold end. The infrared detector was used to continuously measure the corresponding temperature rise process of the central part of the upper surface of the sample. The curve of temperature as a function of time was obtained. The thermal diffusivity was obtained from the equation: α= 0.1388 × d2 / t1/2, which d was the thickness of the samplet1/2 was the semi-temperature rising time which can be read in Figure 1. The specific heat of the sample was obtained by DSC 204 F1 Phoenix (Netzsch, Germany), the density was measured using a real density meter with exhaust method. Thermal conductivity λ (W/mK) was calculated as a multiplication of density (ρ, g/cm3), specific heat (Cp, J/gK), and thermal diffusivity (α, mm2/s). Namely: λ(T)=α(T) × Cp(T) × ρ(T).

2) Thermal conductivity λ (W/mK) was obtained by multiplying the density (ρ, g/cm3), specific heat (Cp, J/gK), and thermal diffusivity (α, mm2/s). We have added the density of samples in Table S1, measured using an exhaust method. 

Finally, the comments and suggestions from the reviewers are highly appreciated.

Reviewer 2 Report

pThe manuscript "Thermally conductive and electrical insulation BNNS/CNF aerogel nano-paper" is interesting to read and shows clearly the improvement of thermal cunductivity with corresponding less increase in electrical conductivity.

The manuscript and the scientific statement are well presented and only minor changes are suggested:

As an aerogel the bulk density of the material would be interesting. I did not see it in the paper, nore the Table S1 in the supplemental information.

Table S1 in the supplemental information had a display issue, i.e. the units where hidden somehow.

For the graphs and the table information on the error in the presented numbers would be advantageous in order to conclude on the significance of a change / difference, especially for the volume resistivity. 

Spedific comments:

Line 14: "For the purpose..." please reformulate

Lines 16-19:In the listed results it seems different samples are compared. It could be helpful for clear overview to stay with the same 2 or three samples and give the corresponding thermal conductivity and resistivity. First its aerogel and blended nanopaper (line 16-17) then its pure CNF nanopaper and aerogel nanopaper (line 18). Please correct in a way suitable for you.

Introduction Line 26-49: The references seem a bit selective, i.e. citing several times a few authors. The background could be introduced somewhat broader by including additional references / review papers of the field. E.g. only one citation from 2011 (ref 16) is given to motivate and highlight particles filled polymers

Line 146: The sacle bars in Fig2 c and b are not readable.

Line 170: Figure 3 b, there is no XRD for the 20 w% - is this due to the fact that it is very similar to the 40 w%?

Line 189: The range of thermal conductivity was increased - What does "range of thermal conductivity" mean, i.e. what are the physical units and what would be an absolute value,e.g. for pure CNF sample to which the "increased range (%)" seems to refer to.

Line 200: "can fully mee the insulation requirements [33]". It would be helpful to get direct information what the criteria are thus to be able to compare measured values directly with criteria. It would furthermore be interesting to give one or two examples of other nano-filled polymers in comparison to discuss in what range the nano-paper is situated with respect to different approaches. 

Author Response

The manuscript "Thermally conductive and electrical insulation BNNS/CNF aerogel nano-paper" is interesting to read and shows clearly the improvement of thermal conductivity with corresponding less increase in electrical conductivity. The manuscript and the scientific statement are well presented and only minor changes are suggested:

Point 1: As an aerogel the bulk density of the material would be interesting. I did not see it in the paper, nor the Table S1 in the supplemental information. Table S1 in the supplemental information had a display issue, i.e. the units where hidden somehow.

Response 1: The density of the materials is important for thermal conductivity. Thermal conductivity λ (W/mK) was obtained by multiplying the density (ρ, g/cm3), specific heat (Cp, J/gK), and thermal diffusivity (α, mm2/s), namely: λ(T)=α(T) × Cp(T) × ρ(T). In this study, the density of the nano-paper was measured via an exhaust method. We have added the density of the sample in Table S1 and have resized the table.

Point 2: For the graphs and the table information on the error in the presented numbers would be advantageous in order to conclude on the significance of a change / difference, especially for the volume resistivity.

Response 2: All results in Figure 5a, 5b and 5d are the averages of the three tests. For Figure 5a, there is only ~±0.01 error, so that the error bars would be too small to be noticed even if they were added; so does the Figure 5b. We managed to add the error bars in Figure 5d, which were based on 3 repeats in the test. Please see the revised Figure 5d with error bars in the revised manuscript. And the errors in Table S1 have already been added.

Specific comments:

Point 1. Line 14: "For the purpose..." please reformulate

Response 1: “For the purpose of comparison” has been changed to “As a comparison”.

Point 2. Lines 16-19: In the listed results it seems different samples are compared. It could be helpful for clear overview to stay with the same 2 or three samples and give the corresponding thermal conductivity and resistivity. First its aerogel and blended nano-paper (line 16-17) then its pure CNF nano-paper and aerogel nano-paper (line 18). Please correct in a way suitable for you.

Response 2: We have corrected this by comparing two same types (nano-paper) samples, i.e., BNNS/CNF aerogel nano-paper and blended nano-paper. Please see the revised manuscript for details.

Point 3. Introduction Line 26-49: The references seem a bit selective, i.e. citing several times a few authors. The background could be introduced somewhat broader by including additional references/review papers of the field. E.g. only one citation from 2011 (ref 16) is given to motivate and highlight particles filled polymers

Response 3: We have added 6 other new references in the introduction.

Point 4. Line 146: The scale bars in Fig2 c and b are not readable. 

Response 4: The scale bars have been adjusted accordingly.

Point 5. Line 170: Figure 3 b, there is no XRD for the 20 w% - is this due to the fact that it is very similar to the 40 w%?

Response 5: The purpose of testing the XRD is to prove the facts that there was no chemical reaction between BNNS and CNF, and the chemical structures of both were preserved. Indeed, the film with BNNS contents of 20 wt% and 40 wt% would have a similar crystal structure. Therefore, only the one at 40% was reported.

Point 6. Line 189: The range of thermal conductivity was increased - What does "range of thermal conductivity" mean, i.e. what are the physical units and what would be an absolute value,e.g. for pure CNF sample to which the "increased range (%)" seems to refer to. 

Response 6: 1) “The range of thermal conductivity” means “the increments of the thermal conductivity”, which was represented by (%) as a unit. 2) The value for the pure CNF nano-paper without BNNS was provided as a reference, which has been included in the revised manuscript.

Point 7. Line 200: "can fully meet the insulation requirements [33]". It would be helpful to get direct information what the criteria are thus to be able to compare measured values directly with criteria. It would furthermore be interesting to give one or two examples of other nano-filled polymers in comparison to discuss in what range the nano-paper is situated with respect to different approaches.

Response 7: 1)According to the reference [39], the criteria required for proper insulation is the volume resistivity (Φ) greater than 109 Ω·cm. We have added the criteria directly in the manuscript. Both the BNNS/CNF nano-paper obtained from different methods can fully meet the insulation requirements, even at low loading or without the BNNS. 2) Feng et al (Feng, C. P.; Bai, L.; Bao, R. Y.; Liu, Z. Y.; Yang, M. B.; Chen, J.; Yang, W. (2018). Electrically insulating POE/BN elastomeric composites with high through-plane thermal conductivity fabricated by two-roll milling and hot compression. Advanced Composites and Hybrid Materials, 1, 160-167.) reported an electrical insulating POE/BN composites, the volume resistivity of the composite was about 1013 Ω·cm, which were lower than what we achieved in this work (1014 Ω·cm). 

Finally, the comments and suggestions from the reviewers are highly appreciated.
